# Craniofacial Defects in Embryos with Homozygous Deletion of *Eftud2* in Their Neural Crest Cells Are Not Rescued by *Trp53* Deletion

**DOI:** 10.3390/ijms23169033

**Published:** 2022-08-12

**Authors:** Marie-Claude Beauchamp, Alexia Boucher, Yanchen Dong, Rachel Aber, Loydie A. Jerome-Majewska

**Affiliations:** 1Research Institute, McGill University Health Centre, Glen Site, Montreal, QC H4A 3J1, Canada; 2Department of Anatomy and Cell Biology, McGill University, Montreal, QC H3A 2B2, Canada; 3Department of Human Genetics, McGill University, Montreal, QC H3A 0G1, Canada; 4Department of Pediatrics, McGill University, Montreal, QC H4A 3J1, Canada

**Keywords:** *Eftud2*, P53, MFDM, spliceosomopathies, neurocristopathies, splicing, neural crest cells, craniofacial

## Abstract

Embryos with homozygous mutation of *Eftud2* in their neural crest cells (*Eftud2^ncc−/−^*) have brain and craniofacial malformations, hyperactivation of the P53-pathway and die before birth. Treatment of *Eftud2^ncc−/−^* embryos with pifithrin-α, a P53-inhibitor, partly improved brain and craniofacial development. To uncover if craniofacial malformations and death were indeed due to P53 hyperactivation we generated embryos with homozygous loss of function mutations in both *Eftud2* and *Trp53* in the neural crest cells. We evaluated the molecular mechanism underlying craniofacial development in pifithrin-α-treated embryos and in *Eftud2*; *Trp53* double homozygous (*Eftud2^ncc−/−^*; *Trp53^ncc−/−^*) mutant embryos. *Eftud2^ncc−/−^* embryos that were treated with pifithrin-α or homozygous mutant for *Trp53* in their neural crest cells showed reduced apoptosis in their neural tube and reduced P53-target activity. Furthermore, although the number of SOX10 positive cranial neural crest cells was increased in embryonic day (E) 9.0 *Eftud2^ncc−/−^*; *Trp53^ncc−/−^* embryos compared to *Eftud2^ncc−/−^* mutants, brain and craniofacial development, and survival were not improved in double mutant embryos. Furthermore, mis-splicing of both P53-regulated transcripts, *Mdm2* and *Foxm1*, and a P53-independent transcript, *Synj2bp*, was increased in the head of *Eftud2^ncc−/−^*; *Trp53^ncc−/−^* embryos. While levels of *Zmat3*, a P53- regulated splicing factor, was similar to those of wild-type. Altogether, our data indicate that both P53-regulated and P53-independent pathways contribute to craniofacial malformations and death of *Eftud2^ncc−/−^* embryos.

## 1. Introduction

Pathogenic variants in *EFTUD2*, which encodes a GTPase and a core component of the U5 subunit of the spliceosome [1], are responsible for the congenital syndrome Mandibulofacial Dysostosis with Microcephaly (MFDM) (OMIM#610536). MFDM patients have a range of craniofacial abnormalities that include micrognathia, small dysplastic pinnae, malar hypoplasia, hearing loss, microcephaly and developmental delay [2]. Mutant mice carrying a heterozygous loss-of-function mutation in *Eftud2* are viable and fertile, and do not model MFDM [3]. On the other hand, embryos that are homozygous for the same loss of function mutation fail to implant and arrest at the blastocyst stage [3]. However, *Eftud2^ncc−/−^* embryos with *Wnt1-Cre2* mediated deletion of exon 2 of *Eftud2* have severe brain and craniofacial malformations, in the same precursor that is affected in MFDM patients, thereby modelling those head and face defects that are found in patients [4].

RNAseq analysis using heads of *Eftud2^ncc−/−^* embryos, prior to the onset of malformations, revealed increased skipping of exon 3 of *Mdm2*, a master regulator of *Trp53* (P53). Additionally, the levels of nuclear P53 and P53-regulated genes, including *Mdm2*, were increased in *Eftud2^ncc−/−^* embryos and in O9-1 neural crest cells after siRNA mediated knockdown of *Eftud2*. Furthermore, levels of P53 target genes were reduced when full-length *Mdm2* was overexpressed in O9-1 cells with *Eftud2* knockdown, and treatment of pregnant females with pifithrin-α improved craniofacial development in *Eftud2^ncc−/−^* embryos [4]. Therefore, we proposed that craniofacial malformations that are found in *Eftud2^ncc−/−^* embryos was a consequence of the mis-splicing of *Mdm2* leading to hyperactivation of the P53 pathway.

Herein, we show that P53 level and activity are reduced in *Eftud2^ncc−/−^* embryos that were treated with pifithrin-α. These embryos also had reduced apoptosis in their neural tube. We generated *Eftud2^ncc−/−^*; *Trp53^ncc−/−^* double mutant embryos and showed that they had reduced nuclear P53 and P53-activity, no changes in mitosis, and reduced apoptosis in their neural tube. Furthermore, although *Eftud2^ncc−/−^*; *Trp53^ncc−/−^* mutants have more SOX10 positive cranial neural crest cells, when compared to *Eftud2^ncc−/−^* (*Eftud2^ncc−/−^*; *Trp53*
^+/+^) embryos, they are morphologically abnormal from embryonic day (E)9.5 to E14.5 and they die before birth, similar to the *Eftud2^ncc−/−^* mutants. Additionally, *Eftud2^ncc−/−^*; *Trp53^ncc−/−^* embryos have a significant increase in exon-skipping in P53-regulated transcripts, including *Mdm2* and *FoxM1*, and a non-significant increase in levels of the P53-independent transcript *Synj2bp*, when compared to controls or *Eftud2^ncc−/−^* mutants. Our data indicate that P53 hyperactivation contributes to apoptosis in the neural tube and suggests that P53 attenuates mis-splicing in *Eftud2* mutant cells. Notably our study shows that craniofacial malformations and death of *Eftud2^ncc−/−^* mutant embryos are independent of P53-activation.

## 2. Results

### 2.1. Pifithrin-α Reduces Levels of Nuclear P53, P53-Target Genes and Apoptosis in the Neural Tubes of Eftud2^ncc−/−^ Embryos

We previously showed that pifithrin-α treatment from E6.5 – E8.5 partially improves brain and craniofacial development in E9.5 *Eftud2^ncc−/−^* mutants [4]. To determine if reduced P53 is responsible for this rescue, we examined P53 accumulation and activity in these previously generated E9.5 embryos. The levels of the three P53-regulated genes *Ccng1*, *Trp53inp1*, and *Phlda3* which were significantly upregulated in heads of E9.0 *Eftud2^ncc−/−^* embryos [4], were also increased in the E9.5 mutant embryos from vehicle-treated females when compared to their control littermates (Figure 1A–C), although this difference was only significant for *Ccng1* and *Trp53inp1* (Figure 1A,B). In contrast, the levels of these P53-target genes were comparable in the heads of pifithrin-α-treated controls and *Eftud2^ncc−/−^* embryos (Figure 1A–C). We examined nuclear P53 accumulation in the pifithrin-α-treated embryos and found a significant increase of P53 positive nuclei in the neural tube, anterior to rhombomere 4, of the vehicle-treated *Eftud2^ncc−/−^* embryos when compared to the controls (Figure 1D,E), but not in their first pharyngeal arch (Figure 1F,G). In contrast, the number of P53 positive cells in the neural tube or pharyngeal arches of pifithrin-α treated *Eftud2^ncc−/−^* embryos was comparable to those of the controls (Figure 1D–G). These findings show that pifithrin-α treatment abolished increased P53-accumulation and activity in the neural tube of *Eftud2^ncc−/−^* embryos.

The number of dying cells found in the head of E9.5 *Eftud2^ncc−/−^* embryos was significantly increased when compared to the controls [4]. To identify the cell population undergoing apoptosis and to determine if pifithrin-α treatment modulates this increase, we quantified the number of cleaved Caspase-3-positive (apoptotic cells) in the neural tube, anterior to rhombomere 4, and the first pharyngeal arch of vehicle and pifithrin-α treated E9.5 embryos (Figure 1H–K). At this stage, *Eftud2^ncc−/−^* embryos from vehicle treated females exhibited an increase in cleaved Caspase-3 positive cells in their neural tube and first pharyngeal arch, when compared to controls (Figure 1H,J). In contrast, although the proportion of apoptotic cells in both structures was increased in *Eftud2^ncc−/−^* embryos that were treated with pifithrin-α the difference was only significant in the pharyngeal arches (Figure 1H,J). In addition, pifithrin-α treatment did not impact proliferation in the head of *Eftud2^ncc−/−^* mutant embryos (Appendix A). These data indicate that pifithrin-α reduced apoptosis in the neural tube of *Eftud2^ncc−/−^* embryos, but not in the pharyngeal arch. Altogether, our data suggest that the improved craniofacial development found in pifithrin-α treated *Eftud2^ncc−/−^* embryos at E9.5 was due to reduced P53-activity and P53-associated apoptosis in the neural tube.

### 2.2. Removing Two Alleles of Trp53 Decreases P53-Activity and Apoptosis in E10.5 Mutant Embryos

We next used a previously described mutant mouse line carrying *Trp53* conditional mutation [5] and *Wnt1-Cre2* to remove *Trp53* in the neural crest cells and confirm that its loss improves craniofacial development in *Eftud2^ncc−/−^* embryos. We used two different mating schemes to generate embryos with heterozygous or homozygous loss of *Trp53* (*Eftud2^ncc−/−^*; *Trp53^ncc+/−^* or *Eftud2^ncc−/−^*; *Trp53^ncc−/−^* (see Section 4) (Appendix A)) and compared them to *Eftud2^ncc−/−^* embryos that were generated in our previous study [4].

We postulated that craniofacial development would be improved in E10.5 *Eftud2^ncc−/−^*; *Trp53^ncc+/−^* or *Eftud2^ncc−/−^*; *Trp53^ncc−/−^* embryos with reduced P53-levels and activity. Therefore, we first examined the levels of the P53-target genes *Ccng1*, *Trp53inp1*, and *Phlda3* which were significantly increased in E9.0 and E9.5 *Eftud2^ncc−/−^* embryos. At E10.5, the levels of these genes were reduced in *Eftud2^ncc−/−^*; *Trp53^ncc−/−^* when compared to *Eftud2^ncc−/−^*, and comparable to those of the controls, as expected (Figure 2A). Next, we evaluated if nuclear P53 accumulation was reduced in the neural tube, anterior to rhombomere 4, and the first pharyngeal arch of E10.5 embryos with genetic deletion of *Trp53* (Figure 2B,C). We found a non-significant increase in P53 accumulation in the neural tubes of *Eftud2^ncc−/−^* and *Eftud2^ncc−/−^*; *Trp53^ncc+/−^* embryos that was reduced in *Eftud2^ncc−/−^*; *Trp53^ncc−/−^* mutants (Figure 2B). On the other hand, nuclear P53 accumulation in the first pharyngeal arch of *Eftud2^ncc−/−^* embryos was not affected by the *Trp53* genotype (Figure 2C).

We next tested if the loss of P53 impacts apoptosis or proliferation in the head of double mutant embryos. Apoptosis, as detected by nuclear cleaved-Caspase 3, was increased in the neural tubes of *Eftud2^ncc−/−^* and *Eftud2^ncc−/−^*; *Trp53^ncc+/−^* mutant embryos, although the difference was only significant in *Eftud2^ncc−/−^*; *Trp53^ncc+/−^*, when compared to controls (Figure 2D). However, this increase was no longer observed in *Eftud2^ncc−/−^*; *Trp53^ncc−/−^* embryos (Figure 2D). At E10.5, apoptosis was not significantly increased in the first pharyngeal arch of embryos with any of the three mutant genotypes (Figure 2E). Additionally, no significant changes were observed in proliferation in the head of E10.5 mutant embryos with mutations in both *Eftud2* and *Trp53* (Appendix A). Taken together, these data indicate that deletion of a single allele of *Trp53* was not sufficient to reduce P53 accumulation or apoptosis in the neural tube. However, homozygous deletion of both *Eftud2* and *Trp53* did abolish the increase of nuclear P53 and the increase in apoptosis found in the neural tube of *Eftud2^ncc−/−^* embryos. Thus, we conclude that increased P53 activity mediates apoptosis in the neural tube of *Eftud2^ncc−/−^* embryos.

### 2.3. SOX10 Expression in the Neural Crest Was Higher in Eftud2^ncc−/−^; Trp53^ncc−/−^ Mutant Embryos Than in Eftud2^ncc−/−^ Mutants

To determine if reduction of P53 activity and apoptosis in the neural tube results in an increase in the number of post-migratory cranial neural crest cells, we examined expression of SOX10, which is expressed in these cells, in E9.0 embryos. As shown in Figure 3A–C, SOX10-expression was reduced in the frontonasal mass and first pharyngeal arch of *Eftud2^ncc−/−^* mutant embryos (*n* = 4) (Figure 3B) when compared to controls (*n* = 4) (Figure 3A). However, SOX10 immunoreactivity in *Eftud2^ncc−/−^*; *Trp53^ncc−/−^* mutants (*n* = 4) (Figure 3C) was comparable to that of controls. Thus, we concluded that *Eftud2^ncc−/−^*; *Trp53^ncc−/−^* mutant embryos have more SOX10-positive post-migratory neural crest cells than *Eftud2^ncc−/−^* embryos.

### 2.4. Removing Both Alleles of Trp53 Does Not Improve Craniofacial Development in Eftud2^ncc−/−^ Mutants

We next assessed if reduced apoptosis and the increase in SOX10 expression in the neural crest cells led to improved craniofacial development in *Eftud2^ncc−/−^* mutant embryos when two alleles of *Trp53* was deleted. As shown in Figure 4, craniofacial defects similar to those previously described in *Eftud2^ncc−/−^* mutant embryos [4] were observed in *Eftud2^ncc−/−^*; *Trp53^ncc+/−^* and in *Eftud2^ncc−/−^*; *Trp53^ncc−/−^* mutants. Briefly, at E9.5 and E10.5, all mutants had hypoplasia of the midbrain, the frontonasal prominence, and the first and second pharyngeal arches (Figure 4A,B). To quantify changes in these structures, we measured the perimeter of the first pharyngeal arches, and the dorsal/midbrain expanse of the midbrain in E9.5 and E10.5 embryos. The perimeter of the first pharyngeal arch was significantly reduced in *Eftud2^ncc−/−^* mutants at both of these stages, regardless of *Trp53* genotype (Figure 5A,C). Additionally, although the ventral/dorsal expanse of the midbrain of *Eftud2^ncc−/−^* mutant embryos was reduced at E9.5 and E10.5 regardless of *Trp53* genotype, (Figure 5B,D); at E9.5, this difference was not significant when *Eftud2^ncc−/−^*; *Trp53^ncc−/−^* mutants were compared to controls (Figure 5B). At E11.5, the midbrain region was virtually absent, the frontonasal prominence was abnormally shaped, and the maxillary and the mandibular processes were smaller than in controls (Figure 4C). Also, the neural tube was open in *Eftud2^ncc−/−^* (*n* = 3/3), in *Eftud2^ncc−/−^*; *Trp53^ncc+/−^* (*n* = 5/5) and in *Eftud2^ncc−/−^*; *Trp53^ncc−/−^* (*n* = 5/6) mutant embryos (data not shown). In addition, the proportion of E11.5 *Eftud2^ncc−/−^*; *Trp53^ncc+/−^* and *Eftud2^ncc−/−^*; *Trp53^ncc−/−^* embryos that were found alive—with craniofacial abnormalities—or dead was not significantly different from what was found for *Eftud2^ncc−/−^* mutants (Figure 5E).

At E14.5, four of the sixteen *Eftud2^ncc−/−^*; *Trp53^ncc+/−^* embryos were alive, among which one was phenotypically normal (*n* = 7 litters). Alcian blue staining revealed no cartilage defects in the phenotypically normal *Eftud2^ncc−/−^*; *Trp53^ncc+/−^* mutant (Appendix A). In contrast, no phenotypically normal *Eftud2^ncc−/−^*; *Trp53^ncc−/−^* embryo was recovered at this stage (*n* = 3 litters). In fact, only one phenotypically abnormal *Eftud2^ncc−/−^*; *Trp53^ncc−/−^* embryo was recovered alive at E14.5 (*n* = 3 litters) (Figure 4D and Figure 5F).

In phenotypically abnormal *Eftud2^ncc−/−^*; *Trp53^ncc+/−^* embryos (*n* = 3) and in the *Eftud2^ncc−/−^*; *Trp53^ncc−/−^* embryo, cartilage preparations with Alcian blue revealed severe craniofacial defects (Fig.S3). Meckel’s cartilage, which will form the lower jaw, was present and smaller in one *Eftud2^ncc−/−^*; *Trp53^ncc+/−^* embryo but absent in the remaining (*n* = 2/3). This cartilage was not found in the single *Eftud2^ncc−/−^*; *Trp53^ncc−/−^* embryo. The paranasal cartilage, nasal and basal portion of the trabecular plate and the orbital cartilage were missing in all the phenotypically abnormal mutants (*n* = 4/4). The frontal cartilage was abnormally shaped (2/4) or absent (2/4) in the mutant embryos. Similarly, the alas temporalis cartilage was absent in *Eftud2^ncc−/−^*; *Trp53^ncc+/−^* (2/3) and *Eftud2^ncc−/−^*; *Trp53^ncc−/−^* embryos and abnormally shaped in one *Eftud2^ncc−/−^*; *Trp53^ncc+/−^* embryo, while the basitrabecular process was absent in all the morphologically abnormal mutants (4/4). Similarly, cartilages of the chondocranium that are derived from head mesoderm, such as the hypochiasmatic cartilage (4/4) and the acrochordal cartilage were absent (3/4) or abnormal (1/4). In contrast, the occipital arch cartilage was present but abnormally shaped in phenotypically abnormal *Eftud2^ncc−/−^*; *Trp53^ncc+/−^* and *Eftud2^ncc−/−^*; *Trp53^ncc−/−^* embryos (Appendix A). Finally, the external ears did not form in any mutants regardless of *Trp53* status. As such, although we recovered one morphologically normal *Eftud2^ncc−/−^*; *Trp53^ncc+/−^* embryo, our data indicate that removing one or both alleles of *Trp53* was not sufficient to rescue craniofacial defects in most *Eftud2^ncc−/−^* embryos.

### 2.5. Homozygous Deletion of Trp53 Does Not Improve Survival of Eftud2^ncc−/−^ Embryos

We next determined if the survival of *Eftud2^ncc−/−^*; *Trp53^ncc−/−^* mutants was prolonged when compared to *Eftud2^ncc−/−^* mutants. No *Eftud2^ncc−/−^*; *Trp53^ncc−/−^* mutants were found at E18.5 (*n* = 0/32, versus expected 8/32, χ^2^ P = 0.0429). In fact, only one resorption could be genotyped and confirmed as an *Eftud2^ncc−/−^*; *Trp53^ncc−/−^* mutant (*n* = 3 litters). Overall, these data indicate that removing two alleles of *Trp53* in *Eftud2^ncc−/−^* mutant embryos did not significantly improve survival.

### 2.6. The Xbp-1-Associated ER-Stress Pathway Is Not Activated in Eftud2^ncc−/−^ Embryos with Heterozygous or Homozygous Deletion of Trp53

Heterozygous mutation of *EFTUD2* in human cells caused mis-splicing and/or mis-expression of key genes that are involved in ER-stress, including increased *XBP1* splicing [6]. Therefore, we tested if *Xbp1* splicing was increased in E9.5 *Eftud2^ncc−/−^* mutants and *Eftud2^ncc−/−^* mutants with a heterozygous or homozygous deletion of *Trp53*. However, no significant difference was found in the splicing of *Xbp1* (Appendix A) when controls (*n* = 6), *Eftud2^ncc−/−^*; *Trp53^ncc+/−^* (*n* = 4) or *Eftud2^ncc−/−^*; *Trp53^ncc−/−^* (*n* = 5) mutant embryos were compared. This data indicates that the *Xbp-1* associated ER stress pathway is unlikely to be a major contributor to craniofacial defects or death of *Eftud2^ncc−/−^* mutants, regardless of their *Trp53* genotype.

### 2.7. FoxM1, a P53-Target, Is Abnormally Spliced in Eftud2^ncc−/−^ Mutant Embryos

In our previous RNAseq analysis using the heads of morphologically normal E9.0 *Eftud2^ncc−/−^* embryos, we showed that mis-splicing of *Mdm2* contributed to upregulation of P53 [4]. We postulated that mis-splicing of a P53-independent or dependent target contributes to these malformations. Therefore, we further analyzed our previously published RNAseq data and found a significant increase in the skipping of exon 7 (*p* = 8.63 × 10^−6^, FDR = 0.005) of *FoxM1*, which is negatively regulated by P53 [7], in *Eftud2^ncc−/−^* embryos. RT-PCR analysis revealed that a transcript with the predicted skipping of exon 7 was present and enriched in *Eftud2^ncc−/−^* mutants, and not found in controls (Appendix A). Furthermore, at E9.5, mis-splicing of *FoxM1* was significantly increased in *Eftud2^ncc−/−^*; *Trp53^ncc−/−^* embryos when compared to controls or *Eftud2^ncc−/−^* mutants with wild-type *Trp53* (Figure 6A). However, although mis-splicing and expression levels of *FoxM1* was increased at E10.5, the expression of its target *Cdc25b* was non-significantly reduced in the heads of E10.5 *Eftud2^ncc−/−^*; *Trp53^ncc−/−^* embryos (Appendix A).

Thus, we conclude that increased mis-splicing of *FoxM1* was unlikely to be a major contributor to craniofacial defects in *Eftud2^ncc−/−^*; *Trp53^ncc−/−^* embryos.

### 2.8. Mis-Splicing Is Increased in Eftud2^ncc−/−^; Trp53^ncc−/−^ Embryos

Since we found an unexpected increase in mis-splicing of *FoxM1* in *Eftud2^ncc−/−^*; *Trp53^ncc−/−^* mutant embryos, we next evaluated exon-skipping in two transcripts which were mis-spliced in *Eftud2^ncc−/−^* mutants wild-type for *Trp53*. RT-PCR analysis of the heads of mutant and control embryos revealed a significant increase in the proportion of *Mdm2* transcripts missing exon 3 in *Eftud2^ncc−/−^*; *Trp53^ncc+/−^* and in *Eftud2^ncc−/−^*; *Trp53^ncc−/−^* embryos, when compared to controls or *Eftud2^ncc−/−^* that are wild-type for *Trp53* (Figure 6B). To determine if P53-independent targets also showed increased mis-splicing, we next interrogated the list of genes that were alternatively spliced in our previous RNAseq analysis from both E9.0 and E9.5 head embryos [4]. Amongst the transcripts with a FDR < 0.01, an inclusion level difference >0.1 and with an average number of reads close to 100, we selected *Synj2bp* for further validation. We designed primers flanking the skipped exon and used RT-PCR to quantify the ratio of the *Synj2bp* full-length and short transcripts in our samples. Although not significant, we found a higher proportion of the shorter transcript without exon 2 of *Synj2bp* in *Eftud2^ncc−/−^*; *Trp53^ncc+/−^* and in *Eftud2^ncc−/−^*; *Trp53^ncc−/−^* mutant heads when compared to controls or *Eftud2^ncc−/−^* embryos wild-type for *Trp53* (Figure 6C). Altogether these data indicate that the presence of wild-type *Trp53* might attenuate mis-splicing in *Eftud2^ncc−/−^* mutant cells. 

### 2.9. Zmat3 Is Not Upregulated in Eftud2^ncc−/−^; Trp53^ncc−/−^ Embryos

ZMAT3 is a P53-regulated RNA splicing factor that regulates exon skipping in a number of transcripts, including exon 3 of *Mdm2* [8,9]. Therefore, we examined the expression of this gene to determine if it is responsible for the increased mis-splicing that was found in *Eftud2^ncc−/−^* mutants with reduced P53 activity. At E9.5 and E10.5, *Zmat3* expression was significantly increased in *Eftud2^ncc−/−^* mutants that are wild-type for *Trp53*, when compared to controls (Figure 6D,E). However, homozygous deletion of *Trp53* attenuated this increase at E9.5 (Figure 6D). Additionally, although levels of *Zmat3* were increased in E10.5 *Eftud2^ncc−/−^*; *Trp53^ncc−/−^* mutants (Figure 6E), this difference was not significant when compared to controls. Thus, we conclude that P53 reduces mis- splicing in *Eftud2^ncc−/−^* mutant cells in a ZMAT3-independent fashion.

## 3. Discussion

Herein, we investigated the contribution of the P53-pathway to craniofacial malformations in *Eftud2^ncc−/−^* mutant embryos. We show that we can reduce P53 levels and activity in the neural tube by treating *Eftud2^ncc−/−^* embryos with pifithrin-α or using *Wnt1-Cre2* to delete both alleles of *Trp53*. Furthermore, although reducing P53-activity attenuated apoptosis in the neural tube of *Eftud2^ncc−/−^* embryos, it did not rescue craniofacial development. We also show that mis-splicing is increased in *Eftud2^ncc−/−^*; *Trp53^ncc−/−^* mutants and that this increase was not associated with a significant change in levels of the P53-regulated splicing factor, *Zmat3.* These data indicate that P53-independent pathways contribute to craniofacial defects in *Eftud2^ncc−/−^* mutants and reveal a previously unappreciated role for P53 in regulating mis-splicing.

The increased accumulation and activity of P53 secondary to mutations in genes that are involved in a diverse array of cellular processes, including splicing, contribute to malformations in animal models of developmental syndromes [10,11]. In a few cases, increased P53 activity was also found in clinical samples from patients with developmental syndromes [11]. More specifically, mouse models of neurocristopathies have overactivation of the P53 pathway and reducing the levels of P53 improve and in some cases rescue malformations in these models [12,13]. A similar partial rescue was found when *Eftud2* was mutated in zebrafish [14,15]. Based on our previous findings that pifithrin-α improved neural tube and craniofacial development in *Eftud2^ncc−/−^* embryos we postulated that reducing P53 in mouse would lead to a similar outcome [4].

In the present study, we show that P53-activity was increased in the heads of E9.5 *Eftud2^ncc−/−^* embryos and that this increase was attenuated by pifithrin-α. More specifically, we show that *Eftud2^ncc−/−^* mutant embryos at this stage had increased nuclear P53 accumulation and apoptosis in the neural tube, but not the first pharyngeal arch. Therefore, and not surprisingly, treating these mutant embryos with pifithrin-α led to reduced apoptosis in the neural tube and not in the first pharyngeal arch. We previously showed that the perimeter of the first pharyngeal arch of pifithrin-α treated *Eftud2^ncc−/−^* embryos is larger than those of vehicle-treated mutants [4]. Therefore, it is surprising that we found a significant increase in apoptosis in pharyngeal arches of embryos that were exposed to pifithrin-α from E6.5 to E8.5. Although this data suggest that P53-independent activity leads to the death of *Eftud2* mutant cells in the pharyngeal arches, further work is needed to decipher how a loss of *Eftud2* results in abnormal development of the first arch and its derivatives, including Meckel’s cartilage. Although first identified as a P53 inhibitor, pifithrin-α can also have P53-independent activity [16]. In fact, its mechanism of action remains unclear and its specific effect on P53 may vary in different cell types. For instance, pifithrin-α was associated with cell-type specific phosphorylation of P53 and the differential expression of its target genes [16]. Moreover, it was reported to reduce intracellular reactive oxygen species (ROS) independently of P53 [16]. To rule out the contribution of a P53-independent mechanisms to the partial rescue of craniofacial development that were found in pifithrin-α-treated *Eftud2^ncc−/−^* embryos, we used a genetic approach and deleted *Trp53* in *Eftud2* mutant cells.

When we used the *Wnt1-Cre2* transgenic mouse line to remove both alleles of *Eftud2* and one allele of *Trp53* (*Eftud2^ncc−/−^*; *Trp53^ncc+/−^* embryos), there was still an increase in nuclear P53 accumulation and in P53 activity. Additionally, most *Eftud2^ncc−/−^*; *Trp53^nc+/−^* embryos resembled *Eftud2^ncc−/−^* mutants, except for one morphologically normal embryo that was recovered at E14.5. Therefore, we cannot completely rule out the possibility that reducing the levels of P53 can prevent craniofacial defects in *Eftud2^ncc−/−^* mutants. However, this is likely to be a very rare event. In fact, when all stages were combined, no additional morphologically normal *Eftud2^ncc−/−^*; *Trp53^ncc+/−^* (*n* = 42) or *Eftud2^ncc−/−^*; *Trp53^ncc−/−^* (*n* = 35) embryos were found. Unfortunately, molecular studies could not be performed on this embryo as the litter was collected for cartilage preparations.

Since *Eftud2* and *Trp53* both map to chromosome 11 and are only 33cM apart, it was difficult to generate both *Eftud2^ncc−/−^* and *Eftud2^ncc−/−^*; *Trp53^ncc−/−^* embryos from the same cross. Therefore, we generated *Eftud2^ncc−/−^*; *Trp53^ncc+/−^* and *Eftud2^ncc−/−^*; *Trp53^ncc−/−^* embryos in separate crosses and compared the morphological and the molecular defects that were found to those that we previously described in *Eftud2^ncc−/−^* embryos. Our current studies show that P53 accumulation and activity were reduced in *Eftud2^ncc−/−^*; *Trp53^ncc−/−^* mutant embryos, suggesting that the P53-pathway was efficiently inhibited. Furthermore, as was found for pifithrin-α treated *Eftud2^ncc−/−^* mutants, apoptosis was significantly reduced in the neural tube of double homozygous mutant embryos. However, in double homozygous mutant embryos the onset of craniofacial malformations, and the size of the pharyngeal arches and the midbrain, were no different from *Eftud2^ncc−/−^* embryos wild type for *Trp53*. Nor did a loss of P53 extend the life of *Eftud2^ncc−/−^* embryos. Altogether, these data indicate that the malformations found in *Eftud2^ncc−/−^* embryos are predominantly P53-independent. In the future, we will identify the pathways that are responsible for these malformations.

Nonetheless, our study suggests that the reduced apoptosis in the neural tube most likely leads to an increase in the number of neural crest cells in *Eftud2^ncc−/−^*; *Trp53^ncc−/−^* embryos. Supporting this, we found reduced SOX10 expression in E9.0 *Eftud2^ncc−/−^* mutants that were morphologically similar to controls. Moreover, removing both copies of *Trp53* resulted in an increase of SOX10 expression in the head of *Eftud2^ncc−/−^*; *Trp53^ncc−/−^* embryos. Using the ROSA26R reporter, we previously showed that the proportion of Cre-expressing cells was similar in control and E9.0 *Eftud2^ncc−/−^* mutant embryos, suggesting that these embryos have a similar number of neural crest cells [4]. Our results from this study indicate that at this stage, *Eftud2^ncc−/−^* mutant cranial neural crest cells have attenuated expression of SOX10, a marker of post-migratory neural crest cells. Thus, we propose that increased P53 activity in the cranial neural crest cells of E9.0 *Eftud2^ncc−/−^* mutants as they exit the neural tube leads to the abnormal expression of proteins such as SOX10, important for their survival and patterning in the pharyngeal arches. Since removing P53 increases expression of SOX10 in *Eftud2^ncc−/−^* mutant cells but does not rescue craniofacial development, we further postulate that SOX10 is not sufficient to protect *Eftud2^ncc−/−^* mutant neural crest cells in the first pharyngeal arch from undergoing cell death at E9.5. However, although we assumed that the number of surviving neural crest cells is insufficient to rescue brain and craniofacial development, we cannot rule out the possibility that surviving neural crest cells cannot differentiate and form the cartilage and bones of the head and face. Furthermore, although we did not see increased splicing of *Xbp1* suggesting that the IRE1α pathway was not activated, we cannot exclude that one of the remaining two- arms of the ER-stress pathway is activated and contributes to Caspase-3 independent cell death of *Eftud2^ncc−/−^* mutant cells.

We next examined our RNAseq dataset for additional pathways/transcripts which may contribute to defects. We examined a role for *FoxM1*, a transcription factor which is essential for G2 progression into mitosis [17], that was mis-spliced in these mutants. *FoxM1* is required for embryonic survival and normal craniofacial development [18], and is expressed in the head and the pharyngeal arches of E10.5 embryos [19]. However, its expression and that of its downstream target, *Cdc25b*, was not significantly changed in mutant embryos, ruling it out as a major contributor to craniofacial defects found in *Eftud2^ncc−/−^* embryos that were wild-type or mutant for *Trp53.*

Surprisingly, the mis-splicing of at least three-transcripts: *Foxm1*, *Mdm2* and *Snynj2b* was increased in *Eftud2^ncc−/−^*; *Trp53^ncc−/−^* embryos, suggesting that wild-type P53 activity tampered mis-splicing. Therefore, we examined the expression of *Zmat3* which encodes for an RNA binding protein that is regulated by P53 [9]. We hypothesized that the expression of this splicing factor would be further increased in *Eftud2^ncc−/−^*; *Trp53^ncc−/−^* embryos, thus explaining the increased exon skipping that was found. However, *Zmat3* overexpression was completely abrogated when *Trp53* was deleted, consistent with the observation that it is regulated by P53 [9]. We propose that one, or more, yet to be identified splicing factors are negatively regulated by P53 and are over expressed in *Eftud2^ncc−/−^*; *Trp53^ncc−/−^* embryos.

Identifying this gene(s) will be essential for determining how P53 attenuates splicing of *Foxm1*, *Mdm2* and *Snynj2b* in *Eftud2* mutant cells. Further RNAseq analysis using the heads of double mutant embryos will be needed to confirm a global increase in mis-splicing.

## 4. Materials and Methods

### 4.1. Mouse Lines

All the procedures and experiments were performed according to the guidelines of the Canadian Council on Animal Care and approved by the Animal Care Committee of the Montreal Children’s Hospital. Wild-type CD1 mice (strain code 022) were purchased from Charles Rivers (Laval, QC, Canada) and wild-type C57Bl/6 mice (stock #000664) were purchased from Jackson Laboratories (Augusta, ME, USA). *Wnt1-Cre2* mice on the 129S4 genetic background were purchased from Jackson’s laboratory (stock# 022137). These *Wnt1-Cre2* transgenic mice express Cre recombinase under the control of the mouse *Wnt1*, wingless-related MMTV integration site 1, promoter and enhancer [20]. The *Trp53^tm1brn^* mouse line on the C57BL/6 genetic background with *loxP* sites flanking exons 2–10 of the *Trp53* gene was purchased from Jackson’s laboratory (*Trp53^loxP/+^*) (stock# 008462) [5]. The generation and the characterization of the conditional *Eftud2^em2Lajm^* (*Eftud2^loxp/+^*) line on the inbred C57BL/6 and mixed CD1 genetic backgrounds, and the *Eftud2^em1Lajm^* (*Eftud2^+/−^*) exon 2 deletion line on a mixed CD1 genetic background were described previously [3,4].

### 4.2. Pifithrin-α Treatment

Pregnant females from mating between *Eftud2^loxP/loxP^* and *Eftud2^+/−^*; *Wnt1- Cre2^tg/+^* mice were injected with 2.2mg/kg pifithrin-α (Sigma-Aldrich, Saint-Louis, Missouri, USA) or 2% DMSO/PBS (vehicle) daily through intra-peritoneal injection, starting at E6.5 until E8.5. The embryos that were collected at E9.5 were previously generated [4,12].

### 4.3. Generation of Mutation in Eftud2 and Trp53 in Neural Crest Cell-Specific

*Trp53* and *Eftud2* both map to mouse chromosome 11 and are approximately 33cM apart. Since these two genes show non-mendelian segregation (Appendix A), *Eftud2^ncc−/−^*; *Trp53^ncc+/−^* (*Eftud2^loxP/−^*; *Trp53^loxP/+^*; *Wnt1-Cre2^tg/+^*) embryos were generated by mating *Eftud2^loxP/+^*; *Trp53^loxp/loxp^* and *Eftud2^+/−^*; *Wnt1-Cre2^tg/+^* mice. We generated double homozygous mutant embryos (*Eftud2^loxP/−^*; *Trp53^loxp/loxp^*; *Wnt1-Cre2^tg/+^*) by mating *Eftud2^loxP/loxP^*; *Trp53^loxp/loxp^* mice to *Eftud2^+/−^*; *Trp53^loxP/loxP^*; *Wnt1-Cre2^tg/+^* mice. *Eftud2^ncc−/−^* (*Eftud2^loxP/−^*; *Wnt1-Cre2* ^tg/+^) mutant embryos that were generated as previously described were used for comparison [4]. From these matings, we used the embryos that did not carry the *Wnt1-Cre2* transgene as controls.

Cre recombinase activity was reported in the male germline of *Wnt1-Cre2* mice on the 129S4 background [21]. Since the embryos carrying homozygous mutation of *Eftud2* in all cells arrest pre-implantation, [3] we did not recover *Eftud2^ncc−/−^* embryos with germline homozygous deletion of *Eftud2*. All embryos that were analyzed in this study were on a mixed genetic background following the multiple crosses that were needed to generate double homozygous mutants.

### 4.4. Collection of Embryos

For embryo collection, the day that a vaginal plug was seen was considered embryonic day 0.5 (E0.5). On the day of dissection, the embryos were removed from their extraembryonic membranes; for stages E8.5 to E10.5, the number of somites was counted under light microscope (Leica MZ6 Infinity1 stereomicroscope). The embryos were fixed in 1 or 4% paraformaldehyde at 4 °C overnight (unless otherwise stated), washed in PBS and kept at 4 °C. The yolk sacs were collected and used for genomic DNA extraction for genotyping.

### 4.5. Cartilage Preparation

To evaluate cartilage formation, embryos were stained with Alcian Blue as previously described [4]. BABB-cleared embryos were visualized under a light microscope (Leica MZ6 Infinity1 stereomicroscope).

### 4.6. Preparation of Embryos for Embedding and Histology

The dissected embryos were fixed in 1% paraformaldehyde overnight. For cryo-embedding, the fixed embryos were first cryoprotected in 30% sucrose overnight, embedded in cryomatrix and sectioned at 10 μm thickness for immunohistochemistry and immunofluorescence.

### 4.7. Immunohistochemistry (IHC) and Immunofluorescence (IF)

Cleaved Caspase-3 (1:250, cat#9661T, Cell Signaling, NEB, Whitby, Ontario, Canada), P53 (1:250, cat#2524, Cell Signaling, NEB, Whitby, ON, Canada), and Phosphohistone H3 (Ser10) (1:200, cat#06-570, Sigma-Aldrich, Saint-Louis, MO, USA) primary antibodies were used. For IHC, an avidin/biotin-peroxidase based system was used (VECTASTAIN^®^ Elite ABC HRP Kit: PK-620000, Vector Laboratories, Newark, CA, USA) containing biotinylated universal (anti-mouse/rabbit IgG) secondary antibody and visualized with DAB (Vector Laboratories). After rinsing with water, the slides were counterstained with Nuclear Fast Red before mounting with an aqueous mounting medium. For quantification of colorimetric signal, particle analysis on Image J was used. 2 to 3 sections per embryo were imaged. To count the number of total cells, the sections after the ones that were used for IHC was mounted with DAPI and particle analysis on Image J was used to determine the total number of cells in the head. The percentage of positive cells were determined as follows: the number of P53 or cleaved Caspase-3-positive cells/number of DAPI-positive cells in the head X100, and plotted using GraphPad (Prism).

For wholemount immunofluorescence, embryos were fixed in 1%PFA overnight, washed with PBS and incubated with SOX10 antibody (1:100, cat#78330S, Cell Signaling, NEB, Whitby, ON, Canada) in blocking buffer (1% BSA, 5% serum, 0.3% TritonX100) overnight at 4 °C. After washing with PBS, embryos were incubated with goat anti-rabbit Alexa 568 (1:500, #A11011, ThermoFisher, Waltham, MA, USA) in blocking buffer overnight with DAPI (1 μg/mL). On the next day, the embryos were washed, and images were captured on Leica microsystem (model DM6000B) and Leica camera (model DFC 450). We thank Dr. Colin Dinsmore for the protocol.

### 4.8. RNA Isolation for RT-qPCR

RNA extraction and RT-qPCR analysis was performed as previously published [4]. The primers used are listed in Table 1.

### 4.9. Primers Used for Splicing Analysis

For *Mdm2*, *FoxM1* and *Synj2bp* splicing analysis, cDNA was amplified with a RT-PCR program that included a hot start at 95 °C for 5 min, followed by 35 cycles of a denaturation step at 95 °C for 10 s, an annealing step at 55 °C for 30 s, an extension step at 72 °C for 45 s with a final extension step at 72 °C for 10 min. The products were visualized on a 2% agarose gel. The primers that were used are as follow: *Mdm2* Forward (exon2-6/7): GATCACCGCGCTTCTCCTGC, *Mdm2* Reverse (exon2-6/7): GATGTGCCAGAGTCTTGCTG [22]; *FoxM1* forward: CTTTAAGCACATTGCCAAGC.

*FoxM1* reverse: GGTTCTGTGGCAGGAAGC; *Synjbp2* forward: CGGAGGAAGAGATCAACCTG, *Synjbp2* reverse: TATCTCACGAAGGCCCAAAC.

The intensity of the bands was quantified using Image J and the ratios of spliced vs. non-spliced were plotted by GraphPad Prism.

### 4.10. ER Stress

To determine if the unfolded protein response pathway is activated in *Eftud2^ncc−/−^*; *Trp53^ncc+/−^* or *Eftud2^ncc−/−^*; *Trp53^ncc−/−^* embryos, we used primers that were previously designed to amplify a 350 bp portion of *Xbp1* (Forward primer: GATCCTGACGAGGTTCCAGA and Reverse primer: GGTCCCCACTGACAGAGAAA), followed by digestion of the amplicon with *Pst1* restriction enzyme, as previously described [23]. Since *Pst1* does not cut the amplicon in the absence of ER stress, the presence of ER stress is indicated by the presence of a 240 bp and a 110 bp bands.

### 4.11. Statistical Analysis

Two-tailed non-parametric Mann-Whitney *t*-test analysis was performed using Excel and Prism Software. ANOVA test analysis followed by Tukey’s post-test to compare all pairs of columns and a Chi-square test were performed using Prism. Significant p-values are represented as * *p* < 0.05, ** *p* < 0.01 and *** *p* < 0.001. All unique/stable reagents that were generated in this study are available from the Lead Contact with a completed Materials Transfer Agreement.

## Figures and Tables

**Figure 1 ijms-23-09033-f001:**
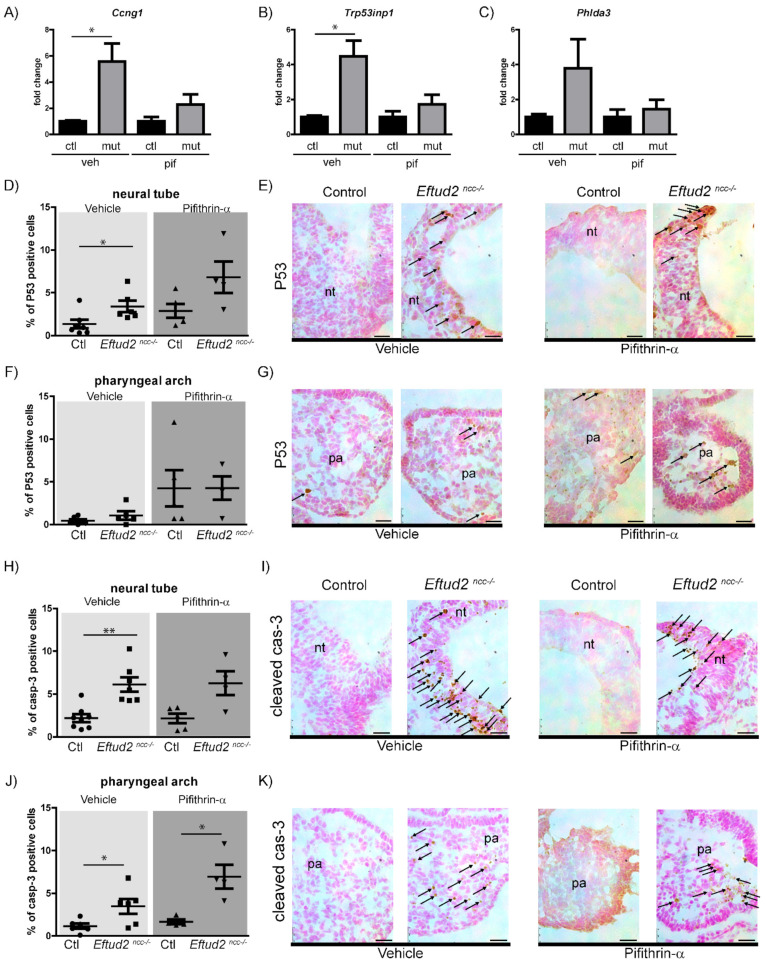
Treatment of *Eftud2^ncc−/−^* embryos with pifithrin-α from E6.5–E8.5 reduces P53-activity, nuclear P53 accumulation, and apoptosis. (**A**–**C**) RT-qPCR analysis revealed significant increases in the levels of (**A**) *Ccng1* and (**B**) *Trp53inp1*, and a non-significant increase in the level of (**C**) *Phlda3* in the heads of *Eftud2^ncc−/−^* embryos (mut, *n* = 3) treated with vehicle (veh), when compared to the controls (ctl, *n* = 3). In pifithrin-α (pif)-treated mutants (mut. *n* = 6), levels of these genes are similar to the controls (ctl, *n* = 4). The Y-axis indicates fold change over the control, the error bars represent SEM, * *p* < 0.05 by *t*-test. (**D**,**F**) Quantification of nuclear P53-positive cells in the neural tube (**D**) and first pharyngeal arch (**F**) of the vehicle and the pifithrin-α treated control (*n* = 7 veh, *n* = 5, pif) and *Eftud2^ncc−/−^* embryos (*n* = 6 veh, *n* = 4, pif). The percentage of P53-positive nuclei was significantly increased in the neural tube of the vehicle treated mutant embryos when compared to the controls. Each dot represents the average percentage of positive cells in a single embryo (* *p* < 0.01 by *t*-test). (**E**,**G**) Representative images of P53 nuclear staining in vehicle (left) or pifithrin-α (right)-treated embryos. Sections were counterstained with nuclear fast red (red), P53-staining is in brown. The arrows indicate P53 positive cells. (**H**) Quantification of cleaved caspase-3-positive cells showing significant increase in the percentage of apoptotic cells in the neural tube of vehicle-treated *Eftud2^ncc−/−^* embryos (*n* = 7) and (**J**) in the first pharyngeal arch of pifithrin-α-treated *Eftud2^ncc−/−^* embryos (*n* = 4), when compared to the control (*n* = 5, veh, *n* = 5, pif). Each dot represents the average percentage of positive cells in a single embryo (* *p* < 0.05, ** *p* < 0.01 by *t*-test). (**I**,**K**) Representative images of cleaved Caspase-3 staining of vehicle or pifithrin-α treated embryos. Arrows indicate cleaved Caspase-3 positive cells. (Genotypes of the control embryos included *Eftud2^loxp/−^* or *Eftud2^loxp/+^*). Scale bar = 25 µm. nt = neural tube, pa = pharyngeal arch.

**Figure 2 ijms-23-09033-f002:**
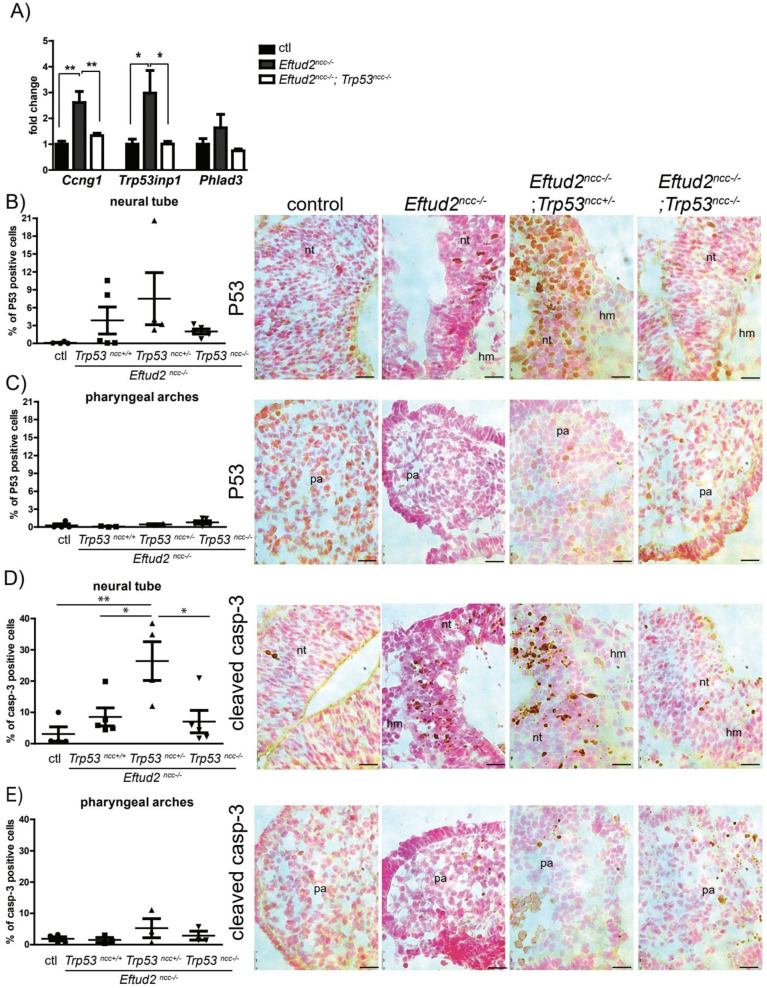
*Eftud2^ncc−/−^*; *Trp53^ncc−/−^* embryos have decreased P53 activity and apoptosis at E10.5. (**A**) RT-qPCR analysis showed that levels of *Ccng1*, *Trp53inp1* and *Phlda3* were significantly increased in the heads of *Eftud2^ncc−/−^* embryos when compared to controls or *Eftud2^ncc−/−^*; *Trp53^ncc−/−^* mutants. Controls (ctl; *n* = 4), *Eftud2^ncc−/−^* (*n* = 4) and *Eftud2^ncc−/−^*; *Trp53^ncc−/−^* (*n* = 6). The Errors bars represent SEM. (**B**) Quantification of the percentage of P53 positive cells showed an increase in the neural tube of *Eftud2^ncc−/−^* and *Eftud2^ncc−/−^*; *Trp53^ncc+/−^* mutant E10.5 embryos, compared to controls or *Eftud2^ncc−/−^*; *Trp53^ncc−/−^* mutants, (**C**) but not in the pharyngeal arch. Each dot represents average percentage of positive cells in an embryo. The right panels show representative images of P53 nuclear staining by immunohistochemistry. The sections were counterstained with nuclear fast red (red), P53-staining is in brown. (**D**) Quantification of the percentage of cleaved Caspase-3-positive cells revealed a significant increase in the neural tube of *Eftud2^ncc−/−^*; *Trp53^ncc+/−^* mutant embryos compared to controls, *Eftud2^ncc−/−^* or *Eftud2^ncc−/−^*; *Trp53^ncc−/−^* mutants, (**E**) but not in the pharyngeal arch. Each dot represents the average of the percentage of positive cells in an embryo. The right panels show representative images of cleaved Caspase-3-positive cells by immunohistochemistry. (ctl *n* = 4: genotypes of control embryos included *Eftud2^loxp/−^*; *Trp53^loxp/+^* or *Eftud2^loxp/+^*; *Trp53^loxp/loxp^*), *Eftud2^ncc−/−^ n* = 5, *Eftud2^ncc−/−^*; *Trp53^ncc+/−^ n* = 4, *Eftud2^ncc−/−^*; *Trp53^ncc−/−^ n* = 5). * *p* < 0.05, ** *p* < 0.01 by ANOVA. Scale bar = 25 µm. nt = neural tube, hm = head mesenchyme, pa = pharyngeal arch.

**Figure 3 ijms-23-09033-f003:**
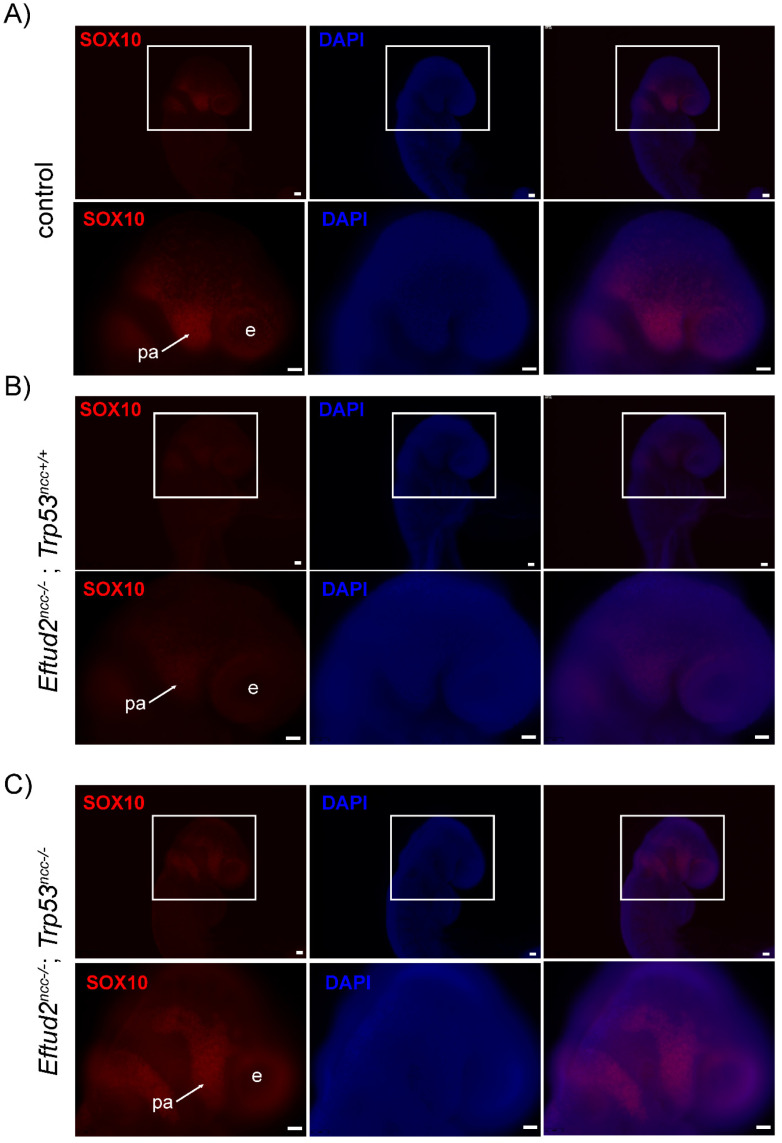
SOX10 expression is higher in *Eftud2^ncc−/−^*; *Trp53^ncc−/−^* embryos than in *Eftud2^ncc−/−^* and similar to controls, at E9.0. Representative images of the head of embryos after wholemount immunofluorescence staining with an antibody to SOX10 (red) or with DAPI (blue) in (**A**) control (*n* = 4), (**B**) *Eftud2^ncc−/−^*; *Trp53^ncc+/+^* (*n* = 4), and (**C**) *Eftud2^ncc−/−^*; *Trp53^ncc−/−^* embryos (*n* = 4). (Genotypes of control embryos included *Eftud2^loxp/−^*; *Trp53^loxp/+^* or *Eftud2^loxp/+^*; *Trp53^loxp/loxp^*). (**A**) In the control embryos, SOX10 is present in post-migratory neural crest cells in and around the eye and in the pharyngeal arches. (**B**) Reduced SOX10 staining is seen in *Eftud2^ncc−/−^*; *Trp53^ncc+/+^* embryos. (**C**) In *Eftud2^ncc−/−^*; *Trp53^ncc−/−^* embryos staining for SOX10 is similar to that seen in controls. Scale bar = 50 µm. e = eye, pa = pharyngeal arch.

**Figure 4 ijms-23-09033-f004:**
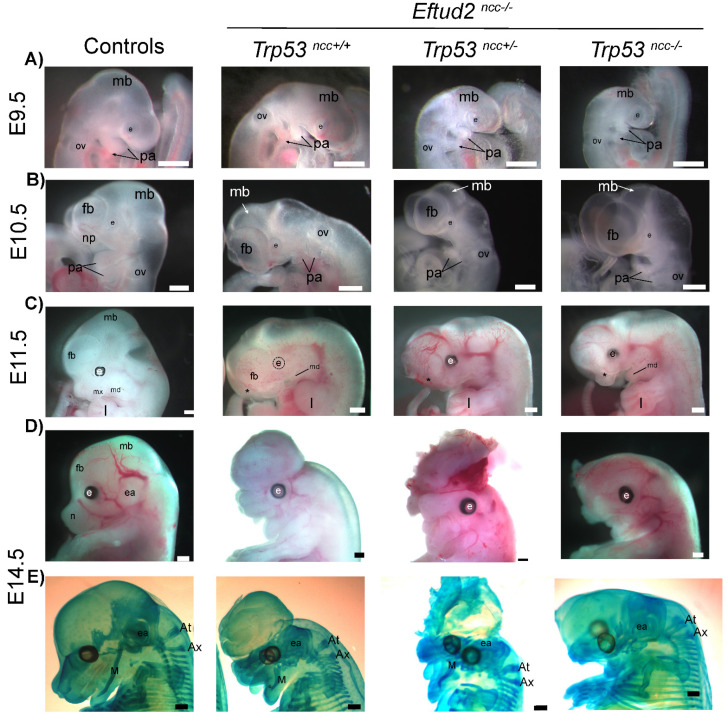
Removing *Trp53* does not improve craniofacial development in *Eftud2^ncc−/−^* mutants. Representative images of control, *Eftud2^ncc−/−^*; *Trp53^ncc+/+^*, *Eftud2^ncc−/−^*; *Trp53^ncc+/−^* and *Eftud2^ncc−/−^*; *Trp53^ncc−/−^* embryos collected at (**A**) E9.5 and (**B**) E10.5 showing the reduced size of the midbrain and the pharyngeal arches. (**C**) The head of mutant embryos have an abnormal curvature; the mandible is reduced and the maxilla and the frontonasal process (stars) was missing in all E11.5 mutant embryos. (**D**) Representative images of E14.5 embryos (**E**) stained with Alcian blue for cartilage analysis showing a hypoplastic (Meckel’s cartilage) and absent cartilage structures in the head of mutants. Scale bar = 500 µm. ov = otic vesicle, mb = midbrain, pa = pharyngeal arch, e = eye, fb = forebrain, np = nasal process, mx = maxillary, md = mandible, l = limb, ea = ear, *n* = nose, M = Meckel’s cartilage, At = atlas, Ax = axis.

**Figure 5 ijms-23-09033-f005:**
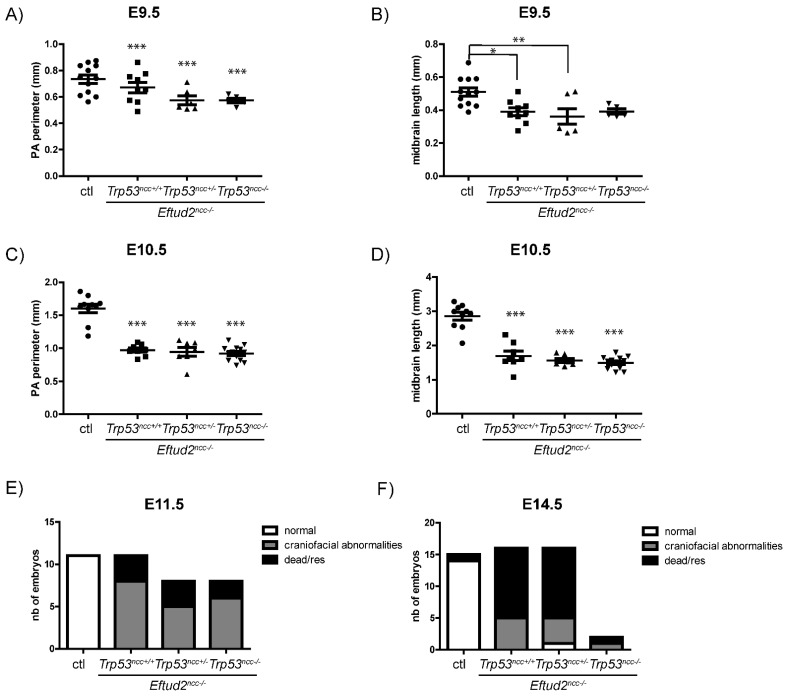
Removing *Trp53* does not improve midbrain and pharyngeal arch growth or improve survival of *Eftud2^ncc−/−^* mutant embryos. Graphs showing that the perimeter of the first pharyngeal arch (PA) was significantly reduced in *Eftud2^ncc−/−^*; *Trp53^ncc+/+^* (*n* = 9, E9.5, *n* = 8, E10.5), *Eftud2^ncc−/−^*; *Trp53^ncc+/−^* (*n* = 6, E9.5, *n* = 7, E10.5) and *Eftud2^ncc−/−^*; *Trp53^ncc−/−^* (*n* = 5, E9.5, *n* = 13, E10.5) embryos at (**A**) E9.5 and (**C**) E10.5 when compared to controls (*n* = 12, E9.5, *n* = 10, E10.5). Graphs showing that the length of the midbrain was reduced in *Eftud2^ncc−/−^*; *Trp53^ncc+/+^* and *Eftud2^ncc−/−^*; *Trp53^ncc+/−^* embryos compared to controls (**B**) at E9.5 (**D**) and also in *Eftud2^ncc−/−^*; *Trp53^ncc−/−^* embryos at E10.5 when compared to controls. * *p* < 0.05, ** *p* < 0.01, *** *p* < 0.001 vs. control by ANOVA. Contingency graphs showing a similar proportion of normal, abnormal and dead or resorbed (res) embryos at (**E**) E11.5 and (**F**) E14.5 within each of these groups: controls (*n* = 11, E11.5, *n* = 15, E14.5), *Eftud2^ncc−/−^*; *Trp53^ncc+/+^* (*n* = 11, E11.5, *n* = 16, E14.5), *Eftud2^ncc−/−^*; *Trp53^ncc+/−^* (*n* = 8, E11.5, *n* = 16, E14.5), and *Eftud2^ncc−/−^*; *Trp53^ncc−/−^* (*n* = 8, E11.5, *n* = 2, E14.5).

**Figure 6 ijms-23-09033-f006:**
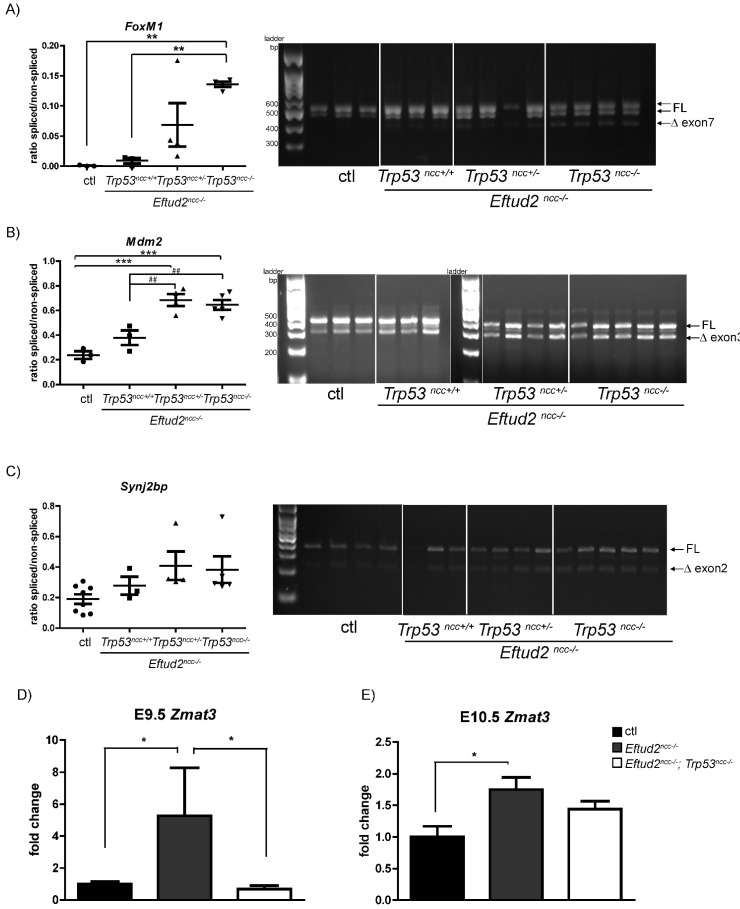
Mis-splicing is increased in *Eftud2^ncc−/−^*; *Trp53^ncc−/−^* embryos. Quantification of the ratio of the skipped-exon containing transcripts vs. the full-length transcripts (FL) (left) and the representative images of gels from RT-PCR analysis (right) showing increased transcripts without (**A**) exon7 of *FoxM1* (controls (*n* = 3), *Eftud2^ncc−/−^*; *Trp53^ncc+/+^* (*n* = 3), *Eftud2^ncc−/−^*; *Trp53^ncc+/−^* (*n* = 4) and *Eftud2^ncc−/−^*; *Trp53^ncc−/−^* (*n* = 4)), (**B**) exon3 of *Mdm2*; (controls (*n* = 3), *Eftud2^ncc−/−^*; *Trp53^ncc+/+^* (*n* = 4), *Eftud2^ncc−/−^*; *Trp53^ncc+/−^* (*n* = 4) and *Eftud2^ncc−/−^*; *Trp53^ncc−/−^* (*n* = 5)) and (**C**) exon2 of *Synj2bp* (controls (*n* = 8), *Eftud2^ncc−/−^*; *Trp53^ncc+/+^* (*n* = 3), *Eftud2^ncc−/−^*; *Trp53^ncc+/−^* (*n* = 4), and *Eftud2^ncc−/−^*; *Trp53^ncc−/−^* (*n* = 5)) in *Eftud2^ncc−/−^*; *Trp53^ncc+/−^* and *Eftud2^ncc−/−^*; *Trp53^ncc−/−^* E9.5 embryos compared to controls or *Eftud2^ncc−/−^*; *Trp53^ncc+/+^*. ** *p* < 0.01, *** *p* < 0.001 vs. control, ^##^
*p* < 0.01 vs. *Eftud2^ncc−/−^* by ANOVA. (**D**) RT-qPCR analysis expressed as fold change over control levels showing increased expression of *Zmat3* in the heads of *Eftud2^ncc−/−^*; *Trp53^ncc+/+^* (*n* = 3) embryos compared to controls (*n* = 6) or *Eftud2^ncc−/−^*; *Trp53^ncc−/−^* (*n* = 5) mutants at E9.5 and (**E**) at E10.5. Controls (*n* = 4), *Eftud2^ncc−/−^*; *Trp53^ncc+/+^* (*n* = 4), and *Eftud2^ncc−/−^*; *Trp53^ncc−/−^* (*n* = 6). (Genotypes of the control embryos at E9.5 included *Eftud2^loxp/−^*; *Trp53^loxp/loxp^* or *Eftud2^loxp/+^*, and at E10.5: *Eftud2^loxp/+^*). The errors bars represent SEM. * *p* < 0.05 by ANOVA.

**Table 1 ijms-23-09033-t001:** Primers used for qRT-PCR.

Genes	Primers
*Ccng1* forward	TTCCAAGATAAGTGGCCGAGA
*Ccng1* reverse	AGTGCGTCCAGACACAATCC
*Trp53inp1* forward	AAGTGGTCCCAGAATGGAAGC
*Trp53inp1* reverse	CTGGGAAGGGCGAAAACTCT
*Phlda3* forward	CATGTCAGCTTCTCTGTCCACTT
*Phlda3* reverse	CTGGTTGGCTCCTTCCATGAT
*Mdm2* forward	TGGAGTCCCGAGTTTCTCTG
*Mdm2* reverse	GATGTGCCAGAGTCTTGCTG
*Sdha* forward	GCTGTGGCCCTGAGAAAGATC
*Sdha* reverse	ATCATGGCCGTCTCTGAAATTC
*B2M* forward	ATGCTATCCAGAAAACCCCTCAA
*B2M* reverse	GCGGGTGGAACTGTGTTACG
*Gapdh* forward	ATGACATCAAGAAGGTCCTG
*Gapdh* reverse	CATACCAGGAAATGAGCTTG
*FoxM1* forward	CTGATTCTCAAAAGACGGAGGC
*FoxM1* reverse	TTGATAATCTTGATTCCGGCTGG
*Cdc25b* forward	TCCGATCCTTACCAGTGAGG
*Cdc25b* reverse	GGGCAGAGCTGGAATGAGG
*Zmat3* forward	TTCCTTTACCTAATCGGCCTTCA
*Zmat3* reverse	TTCCTGCCCAAAAGCCTTCTG

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
