# Peer review of "Craniofacial Defects in Embryos with Homozygous Deletion of *Eftud2* in Their Neural Crest Cells Are Not Rescued by *Trp53* Deletion"

_ijms, 2022, doi:10.3390/ijms23169033_

Round 1
Reviewer 1 Report
This manuscript follows up on the authors’ previous findings that neural crest-specific ablation of Eftud2 in mice leads to craniofacial defects and hyperactivation of the p53 pathway. Here, the authors treated Eftud2 conditional knock-out embryos with a p53 inhibitor and generated double-homozygous mutant Eftud2;Trp53;Wnt1-Cre2embryos. Though both interventions led to decreased apoptosis in the neural tube, neither craniofacial development nor survival improved. The results will likely be of interest to investigators studying alternative RNA splicing in facial development. However, the manuscript in its current state does not support the authors’ conclusions and would benefit from increased experimental details and further analysis of single and double-homozygous mutant embryos.
Major comments:
- Earlier this year the Wnt1-Cre2 allele was shown to exhibit Cre recombinase activity in the male germline in a 129S4 genetic background. The authors should state which genetic background they used here and the breeding scheme used to generate experimental animals (i.e. if the allele was introduced through the male or female germline).
- Information on the number of biological replicates analyzed is often missing from the materials and methods and figure legends and needs to be added.
- For both p53-positive and cleaved caspase-3-positive quantifications in Figures 1 and 2, the percentage of positive cells (as opposed to raw number) should be reported to account for any differences in total cell numbers between treatments or genotypes. Similarly, in Figures 4E and 4F, reporting the findings as percentage of embryos would allow for easier comparison between genotypes.
- For the data shown in Figures 1 and 2, the authors need to clearly state which axial level of the neural tube was analyzed. If the neural tube posterior to rhombomere 4 was analyzed, it would be hard to correlate this data with the craniofacial defects. Similarly, it should be clearly stated which pharyngeal arches were analyzed.
- The authors need to clearly indicate the control genotypes used in each experiment in the text and figures, as it is not clear what those genotypes are from the materials and methods section. Further, the appropriate control genotype for the introduction of the Trp53fl allele in Figure 2 and others would be Eftud2fl/-;Wnt1-Cre2+/Tg. It is present in some analyses but not others with no explanation.
- The authors describe in great detail cartilage defects in the various genotypes. However, most of these defects are virtually impossible to see in Figures 3 and S3. Additional images and alternate angles should be added and/or the discussed structures labeled.
- It is unclear why the authors assessed Sox10 expression in dorsal root ganglia, given their focus on neural crest cells that contribute to craniofacial structures. A similar analysis should be performed assessing Sox10 expression (as a proxy for neural crest cell number) in cranial neural crest cell derivatives.
- Several of the conclusions related to splicing are not supported. For example, the authors state that “Altogether these data indicate that the presence of wildtype Trp53 attenuates mis-splicing in Eftud2ncc-/-mutant cells.” This is an overstatement, as only three transcripts were examined. In order to make this claim and others related to the role of p53 in regulating splicing, the authors would need to perform RNA-seq to look at global splicing changes between genotypes.
Minor comments:
- If known, the Eftud2-null mouse phenotype should be described in the Introduction.
- The catalog number should be indicated for the cleaved caspase-3 antibody.
- It is difficult to identify p53-positive cells and cleaved caspase-3-positive cells in some of the panels in Figures 1 and 2, especially given the variation in background color and counter-stain intensity between sections. This might be alleviated if all (not just a subset of) positive cells were indicated with an arrow. Zoomed-in insets of representative cells would also be helpful.
Author Response
First, we would like to thank reviewers for their reviews and helpful comments on our manuscript. We have addressed the reviewer’s comments below.
Major comments:
- Earlier this year the Wnt1-Cre2 allele was shown to exhibit Cre recombinase activity in the male germline in a 129S4 genetic background. The authors should state which genetic background they used here and the breeding scheme used to generate experimental animals (i.e. if the allele was introduced through the male or female germline).
We thank the reviewer for underlying this point. We previously reported that mouse embryos carrying homozygous mutation of Eftud2 arrest pre-implantation (Beauchamp et al, PLOSOne, 2019). Thus, if germline mutation was introduced by the Wnt1-Cre2 in our mating schemes, we would not have recovered any mutant embryos. This information was added in the manuscript in the introduction and in the material & methods sections where we also specified that the Wnt1-Cre2 allele was introduced using both males and females.
- Information on the number of biological replicates analyzed is often missing from the materials and methods and figure legends and needs to be added.
This was added in all the figure legends.
- For both p53-positive and cleaved caspase-3-positive quantifications in Figures 1 and 2, the percentage of positive cells (as opposed to raw number) should be reported to account for any differences in total cell numbers between treatments or genotypes.
We agree with the reviewer, and we tried using both P53 and cleaved caspase-3 antibodies by immunofluorescence. However, these antibodies did not work well by IF. Thus, we used the consecutive slide used by IHC to quantify the total number of cells in the head of embryos and quantified the % of positive cells as follow: number of P53 or cleaved caspase-3 positive cells/number of DAPI-positive cells X 100. Results were plotted using GraphPad (Prism) and figures 1 and 2 were modified accordingly along with the results section.
Similarly, in Figures 4E and 4F, reporting the findings as percentage of embryos would allow for easier comparison between genotypes.
Although we understand why presenting the data in percentage of embryos can be appealing, we feel that it could also be misleading in some cases. For instance, at E14.5, we recovered significantly less Eftud2ncc-/-; Trp53ncc-/- embryos than expected, suggesting that most embryos have died by this point. Thus, although we are reporting that we found one E14.5 mutant alive and one dead, translating to a 50% survival (Fig 5F), this would be misleading. For that reason, we chose to keep the number of embryos as the y-axis for better clarity.
- For the data shown in Figures 1 and 2, the authors need to clearly state which axial level of the neural tube was analyzed. If the neural tube posterior to rhombomere 4 was analyzed, it would be hard to correlate this data with the craniofacial defects. Similarly, it should be clearly stated which pharyngeal arches were analyzed.
The portion of the neural tube analyzed was anterior to rhombomere 4 and the first pharyngeal arch was analyzed. This is now indicated in the results section.
- The authors need to clearly indicate the control genotypes used in each experiment in the text and figures, as it is not clear what those genotypes are from the materials and methods section. Further, the appropriate control genotype for the introduction of the Trp53fl allele in Figure 2 and others would be Eftud2fl/-;Wnt1-Cre2+/Tg. It is present in some analyses but not others with no explanation.
The genotypes of the controls in each experiment are now clearly indicated in the legends and explained in the Materials & Methods section. We agree with the reviewer that the proper control Eftud2fl/-;Wnt1-Cre2+/Tg is necessary for interpretation of data. We have collected E10.5 Eftud2ncc-/-embryos and included the P53 and cleaved caspase-3 staining in the new Figure 2. We found that the % of P53 positive cells was non-significantly increased in the neural tube compared to controls, but not changed in the pharyngeal arch. Similarly, the % of cleaved caspase-3 positive cells was non-significantly increased in the neural tube compared to controls, but not changed in the pharyngeal arch. These results are now discussed in the manuscript.
- The authors describe in great detail cartilage defects in the various genotypes. However, most of these defects are virtually impossible to see in Figures 3 and S3. Additional images and alternate angles should be added and/or the discussed structures labeled.
We agree with the reviewer and we added new pictures and more labels in Figure S4 for more clarity on the defects represented.
- It is unclear why the authors assessed Sox10 expression in dorsal root ganglia, given their focus on neural crest cells that contribute to craniofacial structures. A similar analysis should be performed assessing Sox10 expression (as a proxy for neural crest cell number) in cranial neural crest cell derivatives.
We have collected embryos at earlier stages (10-13 somites) to assess SOX10 expression in cranial neural crest cells in controls, Eftud2ncc-/-; Trp53ncc+/+ and Eftud2ncc-/-; Trp53ncc-/- (n=4 embryos per genotype). We found that SOX10 positive cells were reduced in all Eftud2ncc-/-; Trp53ncc+/+ embryos compared to controls, specifically around the eye and in the first pharyngeal arch. Interestingly, the Eftud2ncc-/-; Trp53ncc-/- embryos were similar to controls. We have modified our results and discussion section as follow:
Nonetheless, our study suggests that the reduced apoptosis in the neural tube most likely leads to an increase in the number of neural crest cells in Eftud2ncc-/-;Trp53ncc-/- embryos. Supporting this, we found reduced SOX10 expressing cells in E9.0 Eftud2ncc-/- mutants that were morphologically similar to controls. Moreover, removing both copies of Trp53 resulted in an increase in the population of SOX10-positive cells in the head of Eftud2ncc-/-;Trp53ncc-/- embryos. Using the ROSA26R reporter, we showed that the proportion of cre-expressing cells was similar in control and E9.0 Eftud2ncc-/- mutant embryos, suggesting that these embryos have a similar number of neural crest cells [4]. Our results from the current study indicate that at this stage, Eftud2ncc-/- mutant cranial neural crest cells have attenuated expression of SOX10, a marker of post-migratory neural crest cells. Thus, we propose that increased P53 activity in cranial neural crest cells of E9.0 Eftud2ncc-/- mutants as they exit the neural tube leads to abnormal expression of proteins such as SOX10, important for their survival and patterning in the pharyngeal arches. Since removing P53 increases expression of SOX10 in Eftud2ncc-/- mutant cells but does not rescue craniofacial development, we further postulate that SOX10 is not sufficient to protect Eftud2ncc-/- mutant neural crest cells in the first pharyngeal arch from undergoing cell death at E9.5.
- Several of the conclusions related to splicing are not supported. For example, the authors state that “Altogether these data indicate that the presence of wildtype Trp53 attenuates mis-splicing in Eftud2ncc-/-mutant cells.” This is an overstatement, as only three transcripts were examined. In order to make this claim and others related to the role of p53 in regulating splicing, the authors would need to perform RNA-seq to look at global splicing changes between genotypes.
We agree with the reviewer that this was somewhat an overstatement on our part. We have modified the text to tamper our conclusion.
Minor comments:
- If known, the Eftud2-null mouse phenotype should be described in the Introduction.
This information was added in the Introduction.
- The catalog number should be indicated for the cleaved caspase-3 antibody.
This was added.
- It is difficult to identify p53-positive cells and cleaved caspase-3-positive cells in some of the panels in Figures 1 and 2, especially given the variation in background color and counter-stain intensity between sections. This might be alleviated if all (not just a subset of) positive cells were indicated with an arrow. Zoomed-in insets of representative cells would also be helpful.
We have modified our Figures to identify the positive cells with arrows. Also, we added a supplemental Figure S1 showing representative images of higher magnification of P53 and cleaved caspase-3 positive cells.
Reviewer 2 Report
Mandibulofacial dysostosis with microcephaly is a craniofacial spliceosomopathy due to haploinsufficiency of EFTUD2. The authors have developed an animal model for this condition through inactivation of Eftud2 in the mouse neural crest lineage. They have previously shown that these animals have increased P53 activity and increased cell death, as the underlying cause for the brain and craniofacial malformations. Consistent with this observation treatment with pifithrin-a, a P53 inhibitor, rescue the craniofacial and brain phenotypes. In the current study, the authors have generated animals with targeted loss of Eftud2 and Trp53 in neural crest cells to further investigate the role P53. Phenotypic and molecular analyses indicate that in double mutant animals the craniofacial defects are not rescued. The authors propose that craniofacial malformations in Eftud2ncc-/-embryos involve a mechanism independent of P53-overactivation, and pointing to a non-canonical activity of pifithrin-a. This is an interesting follow up to their work published in HMG last year.
-Several data presented in the manuscript are conflicting with previous observations, and this need to be addressed. For example, Figure 1A-C shows that in the mutant embryos Phlda3 is not significantly upregulated as compared to controls, this in contrast to their previous work (see HMG; Fig 6A). Figure 6B, show no significant difference in Mdm2splicing between control and Eftud2ncc-/-;Trp53ncc+/+ mutant, which again is not consistent with previous work (see HMG; Fig 6B).
-In the discussion, the authors state “our data clearly show that reducing P53 chemically or genetically did not significantly improve craniofacial malformations in Eftud2ncc-/-embryos”. This is not completely accurate since they propose that pifithrin-a acts independently of P53 in this context.
-Figure 1H-K indicates that pifithrin-a treatment has opposite effects on neural tube and paryngeal arches reducing and enhancing apoptosis, respectively. How do the authors interpret this increase in caspase-3 positive cells in the pharyngeal arches, which does not correlate with increased P53?
-Figure 5 is missing the appropriate control, Eftud2ncc-/-;Trp53ncc+/+. While the difference is not significant, still the authors claim that there is a higher proportion of neural crest cells surviving in Eftud2ncc-/-;Trp53ncc-/-mutants. This is confusing.
Author Response
Reviewer 2
Mandibulofacial dysostosis with microcephaly is a craniofacial spliceosomopathy due to haploinsufficiency of EFTUD2. The authors have developed an animal model for this condition through inactivation of Eftud2 in the mouse neural crest lineage. They have previously shown that these animals have increased P53 activity and increased cell death, as the underlying cause for the brain and craniofacial malformations. Consistent with this observation treatment with pifithrin-a, a P53 inhibitor, rescue the craniofacial and brain phenotypes. In the current study, the authors have generated animals with targeted loss of Eftud2 and Trp53 in neural crest cells to further investigate the role P53. Phenotypic and molecular analyses indicate that in double mutant animals the craniofacial defects are not rescued. The authors propose that craniofacial malformations in Eftud2ncc-/-embryos involve a mechanism independent of P53-overactivation, and pointing to a non-canonical activity of pifithrin-a. This is an interesting follow up to their work published in HMG last year.
-Several data presented in the manuscript are conflicting with previous observations, and this need to be addressed. For example, Figure 1A-C shows that in the mutant embryos Phlda3 is not significantly upregulated as compared to controls, this in contrast to their previous work (see HMG; Fig 6A).
These data are not conflicting as the previous data published in HMG showing a significant increase in Phlda3 expression were obtained from earlier stage embryos at E9.0, whereas the data presented here were observed in E9.5 embryos. Although not significant, the increased trend in the expression of Phlda3 remains at E9.5.
Figure 6B, show no significant difference in Mdm2splicing between control and Eftud2ncc-/-;Trp53ncc+/+ mutant, which again is not consistent with previous work (see HMG; Fig 6B).
Similarly, the previous data published in HMG showing a significant increase in Mmd2 exon skipping were obtained from earlier stage embryos at E9.0, whereas the data presented here were observed in E9.5 embryos. Although not significant at E9.5, the increased trend in the skipping of exon 2 of Mdm2 remains.
-In the discussion, the authors state “our data clearly show that reducing P53 chemically or genetically did not significantly improve craniofacial malformations in Eftud2ncc-/-embryos”. This is not completely accurate since they propose that pifithrin-a acts independently of P53 in this context.
This was a mistake on our part and we removed that sentence in the current version.
-Figure 1H-K indicates that pifithrin-a treatment has opposite effects on neural tube and paryngeal arches reducing and enhancing apoptosis, respectively. How do the authors interpret this increase in caspase-3 positive cells in the pharyngeal arches, which does not correlate with increased P53?
These data are now discussed as follow:
“We showed that the perimeter of the first pharyngeal arch of pifithrin-a treated Eftud2ncc-/- embryos is larger than those of vehicle-treated mutants [4]. Therefore, it is surprising that we found a significant increase in apoptosis in pharyngeal arches of embryos exposed to pifithrin-a from E6.5 to E8.5. Though this data suggest that P53-independent activity leads to death of Eftud2 mutant cells in the pharyngeal arches, further work is needed to decipher how loss of Eftud2 results in abnormal development of the first arch and its derivatives, including Meckel’s cartilage.”
-Figure 5 is missing the appropriate control, Eftud2ncc-/-;Trp53ncc+/+. While the difference is not significant, still the authors claim that there is a higher proportion of neural crest cells surviving in Eftud2ncc-/-;Trp53ncc-/-mutants. This is confusing.
We agree with the reviewer and this section was modified as described earlier in comment#7 from reviewer 1 below.